# Active Invariant Causal Prediction: Experiment Selection through Stability

**Juan L. Gamella**
Seminar for Statistics
ETH Zurich
Switzerland
gajuan@ethz.ch

**Christina Heinze-Deml**
Seminar for Statistics
ETH Zurich
Switzerland
heinzedeml@stat.math.ethz.ch

## Abstract

A fundamental difficulty of causal learning is that causal models can generally not be fully identified based on observational data only. Interventional data, that is, data originating from different experimental environments, improves identifiability. However, the improvement depends critically on the target and nature of the interventions carried out in each experiment. Since in real applications experiments tend to be costly, there is a need to perform the *right* interventions such that as few as possible are required. In this work we propose a new active learning (i.e. experiment selection) framework (A-ICP) based on Invariant Causal Prediction (ICP) [27]. For general structural causal models, we characterize the effect of interventions on so-called stable sets, a notion introduced by [30]. We leverage these results to propose several intervention selection policies for A-ICP which quickly reveal the direct causes of a response variable in the causal graph while maintaining the error control inherent in ICP. Empirically, we analyze the performance of the proposed policies in both population and finite-regime experiments.

## 1   Introduction

Causal models [24] capture the causal relationships between variables and allow us to predict how a system behaves under interventions or distribution changes. Hence, they are more powerful than probabilistic models, and can be seen as abstractions of more accurate mechanistic or physical models while retaining enough power to answer interventional or counterfactual questions [28]. Therefore, they maintain their predictive power in new, previously unseen environments [13, 35, 31, 30].

The question remains if for systems of interest such models can be learned directly from data. This problem is known in the literature as *causal learning*, and it is to causal models what statistical learning is to probabilistic models. Just like statistical learning, it suffers from the inherent difficulty of determining properties of a distribution from finite-sized samples. Additionally, causal learning is challenged by the fact that, even with full knowledge of the underlying observational distribution, some causal relationships cannot be established and causal models can generally not be fully identified from observational data alone [24]. For causal directed acyclic graph models, this limit of identifiability implies that, from observational data alone, the true graph cannot be distinguished from others that lie in the same Markov equivalence class [38]. Under additional assumptions about the model class and noise distributions, full identifiability is still possible [15, 3, 34, 25, 26]. In the general case, however, identifiability can only be improved by performing interventions (experiments). Examples of such interventions are abundant in the empirical sciences, from gene knockout experiments in biology to chemical compound selection in drug discovery [22]. Since such experiments tend to be costly, there is a need to pick the *right* interventions, in the sense of having to do as few of them as

possible. In the remainder of this section, we review existing work that addresses this problem before summarizing our contributions.

## 1.1 Related work

We use the term *active causal learning* to refer to learning causal models from data while being able to actively perform interventions. In this setting, the goal is to sequentially improve identifiability, as opposed to the classical setting in machine learning [33], where the goal is to sequentially increase prediction accuracy. Existing approaches can be said to fall broadly into two categories: Bayesian and graph-theoretic. The Bayesian approach, pioneered by the works of [37, 21], selects interventions which maximize a Bayesian utility function, generally the mutual information between the graph and the hypothetical sample that the experiment would produce. More recent works build on this approach by considering experiments performed in batches under budget constraints [1] or when expert knowledge is available [20], and apply such framework to learning biological networks [4, 23].

Among the graph-theoretic approaches, [6, 16] give bounds on the number of interventions required for identifiability under different assumptions, and [12, 11] provide intervention selection strategies that aim to orient the maximum number of edges in the graph. There are extensions to several settings, such as when the total number of interventions is limited [8, 7], when there are hidden variables [18] or when interventions carry a cost which must be minimized [17, 19].

Both approaches make different assumptions and suffer from different drawbacks. The Bayesian approach requires exact knowledge of the intervention location and parameters. It is difficult to analyze the impact of misspecified interventions on the choice of experiments and the estimates produced by the methods [39]. Furthermore, it suffers from poor computational scaling [23] and several approximations have to be made even for small graphs [1]. This further complicates giving guarantees on the result. Graph-theoretic approaches are agnostic to the underlying distribution, but generally make two strong assumptions: (1) that the Markov equivalence class has been correctly identified, which is difficult with a limited sample size, and (2) that interventions are perfectly informative (i.e. infinite interventional data).

**Invariant Causal Prediction and intervention stable sets** This work is a first attempt at a new approach which falls into neither of the previous two categories. It relies on Invariant Causal Prediction (ICP) [27], which aims to recover the direct causes $S^*$ of a *response* variable of interest $Y$ from interventional data. The general idea is that the conditional distribution of the response, given its direct causes, remains invariant when intervening on arbitrary variables in the system other than itself. ICP considers the setting where different experimental conditions of a system exist (called *environments*) and an i.i.d. sample of each environment is available. By considering all possible subsets of the predictor variables $X$, ICP then searches for sets of *plausible causal predictors*. These are sets of predictors which, if conditioned on, leave the distribution of the response invariant across the observed environments (see section 3 for the formal definition). This procedure is based on testing the null hypothesis of invariance. Sets considered as plausible causal predictors given the available data are referred to as *accepted sets*, and the set of direct causes of the response (its parents in the causal graph) will be among them with high probability. ICP then returns the intersection of all accepted sets as the estimate $\hat{S}$ of the direct causes. More details are given in Appendix B.

While ICP does not retrieve the full graph, it has some important advantages in the form of guarantees and more flexible assumptions. It requires neither knowledge of the Markov equivalence class, nor about the nature or location of the interventions performed in each environment, except that they must not act on the response. The approach assumes that the noise distribution of the response is independent from the direct causes and invariant across environments. In the general formulation, no further distributional assumptions are made. While such further assumptions can arise from the choice of tests for the invariance of the conditional distribution, non-parametric tests can be chosen [14]. Perhaps most importantly, ICP provides an error control with respect to the estimated causes, namely that with high probability it will not retrieve false positives. While this comes at a loss of power, this work shows that when environments are generated via an appropriate experiment selection strategy, ICP can quickly identify the direct causes while maintaining the aforementioned control. In addition to ICP, we make use of the notion of so-called *intervention stable sets* [30] which relates the invariance properties of a set of predictors to graphical criteria. More details are given in section 2.

## 1.2 Contributions and outline

We propose Active Invariant Causal Prediction (A-ICP), an active causal learning framework based on ICP. Figure 1 shows its core components. In each round, a new intervention target is selected based on the sets accepted by ICP in the previous iteration. Subsequently, the corresponding experiment is performed, yielding a new sample of interventional data[1]. Finally, ICP is run on the updated dataset which yields updated estimates of the accepted sets and the direct causes of $Y$.

Our main contribution lies in the formulation of several policies that choose an intervention target in each round $t$. They are motivated by theoretical results on the invariance properties of sets of predictors (section 2). In section 3, we detail how we combine these results with ICP in an active causal learning setting. We then propose several policies that fit into the A-ICP framework in section 4. While our theoretical results do not require any parametric assumptions on the underlying structural causal model (SCM), we focus on linear SCMs in the empirical evaluation in section 5. In population and finite regime experiments, the proposed policies outperform a random baseline policy across a large range of experimental settings. Finally, we compare A-ICP against ABCD [1] and discuss the observed tradeoffs between error control and power.

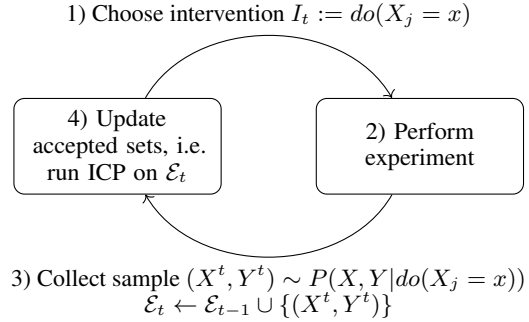

1) Choose intervention $I_t := do(X_j = x)$

4) Update accepted sets, i.e. run ICP on $\mathcal{E}_t$

2) Perform experiment

3) Collect sample $(X^t, Y^t) \sim P(X, Y | do(X_j = x))$
$\mathcal{E}_t \leftarrow \mathcal{E}_{t-1} \cup \{(X^t, Y^t)\}$

Figure 1: Schematic of A-ICP

## 2 Intervention stable sets

We now present the theoretical results that motivate the intervention selection policies in each round $t$ of A-ICP. We use the framework of structural causal models (SCMs) [32, 9, 2]. A SCM consists of (i) a collection of structural assignments that functionally relate each variable in the system to its direct causes and (ii) a joint distribution over the noise variables which are required to be jointly independent. A SCM induces a joint distribution over the variables in the system as well as a graph over the associated vertices (e.g. see Definition 6.2 in [28]). In the following setting, we formalize the assumptions required for the results derived in this section. Importantly, for the results presented in this section, we do not require the SCM to be linear. All proofs can be found in Appendix F.

**Setting 1** (adapted from setting 2 in [30]) Let $X \in \mathcal{X} = \mathcal{X}_1 \times ... \times \mathcal{X}_p$ be predictor variables, $Y \in \mathbb{R}$ a *response* variable and $I = (I_1, ..., I_m) \in \mathcal{I} = \mathcal{I}_1 \times ... \times \mathcal{I}_m$ intervention variables which are unobserved and formalize the interventions present in the collection of intervention environments $\mathcal{E}$. Assume there exists a SCM $\mathcal{S}^{\mathcal{E}}$ over $(I, X, Y)$ that can be represented by a directed acyclic graph $\mathcal{G}(\mathcal{S}^{\mathcal{E}})$, in which the intervention variables are source nodes. Further assume intervention variables do not appear in the structural equation of $Y$, that is, assume there are no interventions on the response. For each $e \in \mathcal{E}$, there is a SCM $\mathcal{S}_e$ over $(I^e, X^e, Y^e)$ such that $\mathcal{G}(\mathcal{S}_e) = \mathcal{G}(\mathcal{S}^{\mathcal{E}})$, in which only the equations with $I^e$ on the right hand side change with respect to $\mathcal{S}^{\mathcal{E}}$. Furthermore, assume that the distribution of $(I^e, X^e, Y^e)$ is absolutely continuous with respect to a product measure that factorizes.

No further assumptions are made on the size or type of the intervention, i.e. they can be do, noise or shift interventions on a single or multiple variables. To simplify notation, let PA($i$), CH($i$) and AN($i$) be the parents, children and ancestors of the variable $X_i$, respectively.

The notion of intervention stable sets, introduced in [30], allows characterizing sets of plausible causal predictors from d-separation relationships in the graph. While stable sets are generally not equivalent to the sets of plausible causal predictors, we here derive theoretical results for them and then analyze under which conditions these apply to the plausible causal predictors (section 3).

**Definition 2.1** (intervention stable set [30]). *Let for any set $S \subseteq \{1, ..., p\}$, $X_S$ be the vector containing all variables $X_k, k \in S$. Given setting 1 and a set of environments $\mathcal{E}$, we call a set*

$S \subseteq \{1, ..., p\}$ intervention stable *under $\mathcal{E}$ if the d-separation $I \perp\!\!\!\perp_{\mathcal{G}} Y \mid X_S$ holds in $\mathcal{G}(\mathcal{S}^{\mathcal{E}})$ for any intervention $I$ which is active in an environment $e \in \mathcal{E}$.*

In other words, a set of predictors is stable if it d-separates the response from all interventions (see Example A.1). In the following, let $\mathbb{S}_{\mathcal{E}}$ denote the collection of sets which are intervention stable under $\mathcal{E}$. The stable sets allow properties of the graph structure and the interventions to be inferred:

**Lemma 1** (intervened parents appear on all intervention stable sets). *Let $\mathcal{E}$ be a set of observed environments and let $j \in PA(Y)$ be directly intervened on in $\mathcal{E}$. Then,*

$$S \subseteq \{1, ..., p\} \text{ is intervention stable} \implies j \in S.$$

**Lemma 2** (sets containing descendants of directly intervened children are unstable). *Let $i \in CH(Y)$ be directly intervened on in $\mathcal{E}$. Then, any set $S \subseteq \{1, ..., p\}$ which contains descendants of $i$ is not intervention stable.*

**Lemma 3** (stability of the empty set). *Let $\mathcal{E}$ be any set of environments. Then,*

$$\emptyset \in \mathbb{S}_{\mathcal{E}} \iff \mathcal{E} \text{ contains no interventions on variables in } AN(Y).$$

That is, the empty set is stable if and only if none of the interventions in $\mathcal{E}$ occurred upstream of $Y$. More structure can be inferred by considering the number of stable sets in which a predictor appears:

**Definition 2.2** (stability ratio). *Given a set of environments $\mathcal{E}$, the* stability ratio *of a variable $i \in \{1, ..., p\}$ is defined as*

$$r_{\mathcal{E}}(i) := \frac{1}{|\mathbb{S}_{\mathcal{E}}|} \sum_{S \in \mathbb{S}_{\mathcal{E}}} \mathbb{1}\{i \in S\},$$

*i.e. the proportion it appears in the intervention stable sets under $\mathcal{E}$.*

From Lemma 1 it follows that parents which are directly intervened on in at least one environment in $\mathcal{E}$ have a stability ratio of 1. Conversely, by Lemma 2 descendants of children directly intervened on in at least one environment have a ratio of 0. Furthermore, the stability ratio of any ancestor, regardless of the interventions, is always larger than one half:

**Proposition 1** (ancestors appear on at least half of all stable sets). *Let $\mathcal{E}$ be any set of observed environments. Then, for any $j \in \{1, ..., p\}$,*

$$r_{\mathcal{E}}(j) < 1/2 \implies j \notin AN(Y).$$

**Corollary 2.1.** *The parents of the response always have a stability ratio of or above $1/2$.*

Note that the converse is not generally true, i.e. variables which are not ancestors can have a stability ratio of or above $1/2$, even after being intervened on. In section 4 we exploit Lemma 1, Lemma 3 and Corollary 2.1 to construct intervention selection policies.

## 3 From stable sets to causal predictors

The results derived in section 2 apply to intervention stable sets. If we are to use these results to construct an intervention selection policy for A-ICP, we need to understand under which conditions they apply directly to the sets of plausible causal predictors.

**Definition 3.1** (plausible causal predictors [27]). *We call a set of variables $S \subseteq \{1, ..., p\}$ plausible causal predictors* under a set of environments $\mathcal{E}$ *if for all $e, f \in \mathcal{E}$ and all $x$*

$$Y^e | X_S^e = x \overset{d}{=} Y^f | X_S^f = x, \tag{1}$$

*i.e. the conditional distribution is the same in all environments. Let $\mathbb{C}_{\mathcal{E}}$ denote the collection of sets which are plausible causal predictors under $\mathcal{E}$.*

Given a collection of environments $\mathcal{E}$, the collection of accepted sets of the ICP algorithm is an estimate of $\mathbb{C}_{\mathcal{E}}$. The following proposition establishes the relationship between intervention stable sets and sets of plausible causal predictors.

**Proposition 2** (intervention stable sets are plausible causal predictors)**.** *Let $\mathcal{E}$ be a set of observed environments. Then, for all intervention stable sets $S \subseteq \{1, ..., p\}$, it holds that $S \in \mathbb{C}_{\mathcal{E}}$.*

While $\mathbb{S}_{\mathcal{E}} \subseteq \mathbb{C}_{\mathcal{E}}$, it is not generally true that $\mathbb{S}_{\mathcal{E}} = \mathbb{C}_{\mathcal{E}}$, even under the faithfulness assumption (see Example A.2). However, when the parameters of the SCM are sampled from a continuous distribution, we conjecture that the set of parameters for which $\mathbb{S}_{\mathcal{E}} \neq \mathbb{C}_{\mathcal{E}}$ has probability zero. We call the assumption that $\mathbb{S}_{\mathcal{E}} = \mathbb{C}_{\mathcal{E}}$ *stability-faithfulness*, and adopt this assumption in the following.

Finally, we make use of the following corollary in A-ICP. In each iteration an intervention target is selected and a sample is collected from the new experimental environment (see Figure 1). Denote by $\mathcal{E}_t = \{e_i : i \in \{1, .., t\}\}$ the set of observed environments at iteration $t$, and assume $\mathcal{E}_t \subseteq \mathcal{E}_{t+1}$.

**Corollary 3.1.** *Let $\mathcal{E}_t, \mathcal{E}_{t+1}$ be sets of observed environments such that $\mathcal{E}_t \subseteq \mathcal{E}_{t+1}$. Then, it follows that if $S$ is not a set of plausible causal predictors under $\mathcal{E}_t$, it is not under $\mathcal{E}_{t+1}$ either.*

## 4 Constructing an active learning policy

Even in the population setting—in the absence of estimation errors—the capacity of ICP to retrieve the parents relies heavily on the informativeness of the environments. For example, if none of the interventions are upstream of the response, the empty set is intervention stable and is returned as estimate of the parents. While [27] gives some sufficient conditions for the identifiability of the true causal parents, it is not entirely clear what an optimal intervention is. If we assume stability-faithfulness, by Lemma 1 we know that, in the absence of estimation errors, a direct intervention on a parent is sufficient for it to appear in the ICP estimate. However, it is not a necessary condition (see Example A.3). As a first approach, we treat direct interventions on the parents as "maximally informative", and the goal of the proposed policies is to pick such interventions. For simplicity and to allow comparison with ABCD [1], we consider single-variable interventions.

**Proposed policies**  To increase the chances of picking a parent of the response as an intervention target, the proposed policies can make use of three strategies:

1. (*Markov strategy*, ("markov")) This strategy selects intervention targets from within the Markov blanket, which contains the parents. Under linearity, in the population setting the Markov blanket can be directly obtained from an ordinary least squares (OLS) regression over all predictors (Appendix E). In the finite regime, we turn to the Lasso [36] to obtain an estimate.

2. (*empty-set strategy*, ("e")) If an observational sample is available, we can test whether the invariance in Eq. (1) holds for the empty set when considering the observational and the interventional sample $e_t$. If it does, by Lemma 3 we know that the latest intervention target is not upstream of the response, and therefore not a parent. We hence discard the target from future interventions.

3. (*ratio strategy*, ("r")) By Corollary 2.1, a variable is not a parent if it appears on less than half of all intervention stable sets. As an estimate we use the accepted sets (computed based on the environments $\mathcal{E}_{t-1}$) and, if a variable appears on less than half of such sets, we do not add it to the pool of possible intervention targets for the current iteration. Note that unlike in (2.), we do not discard it from future interventions. This is important in the finite regime, where parents may for some iterations appear in less than half of all accepted sets due to testing errors.

Furthermore, we exclude identified parents, i.e. variables with a stability ratio of 1, from the pool of possible intervention targets for all of the above strategies. Each strategy narrows down the set of possible intervention targets, and the actual target is then chosen uniformly at random. For multiple-variable interventions we can simply pick $k$ targets instead of one. If a policy combines several of the above strategies, the final set of possible intervention targets is taken as the intersection of the respective strategies' sets. For instance, by combining the *ratio* and the *empty-set* strategies we increase the chance of picking a parent but also exclude non-ancestors which retain a stability ratio above one half after being intervened on. As non-ancestors can have a stability ratio above one half, strategies exploiting the value of the stability ratio (e.g. by intervening on the variable whose value of $r_{\mathcal{E}}(i)$ is closest to $1/2$ in absolute value) are not competitive.

**Algorithm outline** In the A-ICP framework, policies are treated as interchangeable modules which define two functions: `next_intervention` and `first_intervention`. At each iteration of the procedure, the accepted sets (given the current environments $\mathcal{E}_t$) are passed to the policy by calling `next_intervention`, which then returns the next intervention target. Since the accepted sets are not available when selecting the initial intervention, a potentially available (observational) sample can be used to guide a first choice. For example, policies employing the *Markov* strategy compute an estimate of the Markov blanket in this step, and pick a variable within this estimate as the first intervention; other policies pick an intervention at random. For simplicity, we assume the availability of an initial observational sample, even though this is not necessary for all of the proposed strategies.

By Corollary 3.1, at each iteration of A-ICP it suffices for ICP to consider only the sets accepted in the previous iteration, which provides a substantial speed up as not all subsets of predictors need to be re-tested. To account for multiple testing of the accepted sets, we need to apply a correction to the significance level of ICP. Due to the strong dependence between the tests, we use a Bonferroni correction and run ICP at a significance level of $\alpha/T$, where $\alpha$ is the desired overall level and $T$ is the total number of iterations for which A-ICP is run. Hence, the coverage guarantee for the final A-ICP estimate $\hat{S}(\mathcal{E}_T)$ is $P(\hat{S}(\mathcal{E}_T) \subseteq S^*) \geq 1 - \alpha$. Further details and an outline of ICP can be found in Appendix B. More details about the multiple testing correction are given in subsubsection D.2.1 and a sensitivity analysis with respect to the chosen overall significance level $\alpha$ can be found in Appendix D.5. Finally, an analysis of the computational complexity of algorithm 1 is given in Appendix C.

---

**Algorithm 1:** A-ICP

---

**Output :** $\hat{S}(\mathcal{E}_T)$ estimate of the parents of the response
**Input :** `policy` an intervention selection policy,
        $(X^0, Y^0)$ sample from initial environment,
        $T$ number of iterations,
        $\alpha$ overall A-ICP significance level
$\mathcal{E}_0 \leftarrow \{(X^0, Y^0)\}$;
`accepted sets` $\leftarrow$ all sets of predictors;
`next_intervention` $\leftarrow$ `policy.first_intervention`$(\mathcal{E}_0)$;
**for** $t = 1 : T$ **do**
    perform `next_intervention` and collect sample $(X^t, Y^t)$;
    $\mathcal{E}_t \leftarrow \mathcal{E}_{t-1} \cup \{(X^t, Y^t)\}$;
    `accepted sets`, $\hat{S}(\mathcal{E}_t) \leftarrow$ `ICP`$(\mathcal{E}_t,$ `accepted sets`$, \alpha/T)$;   // see Corollary 3.1
    `next_intervention` $\leftarrow$ `policy.next_intervention`(`accepted sets`);
**end**
**return** $\hat{S}(\mathcal{E}_T)$

---

## 5 Experiments

We evaluate policies that use different combinations of the strategies in both the population and finite sample setting, using simulated data from randomly chosen linear SCMs. In addition to averaging over different SCMs, for every SCM, each policy is run a number of times with different random seeds to account for the stochastic component of the policies. Further details about the experimental settings and links to code to reproduce the experiments can be found in Appendix E. Table E.1 summarizes the considered settings and experimental parameters.

### 5.1 Population setting

For the population setting, we evaluate the *Markov* and *ratio* strategies. Population experiments simplify an initial evaluation of the proposed policies. First, since interventions are perfectly informative, the performance of the policies can be compared exclusively in terms of their choice of targets, without worrying about (1) the parameters of the intervention, and (2) how many observations must be allocated to the experiment, neither of which are trivial problems. Second, we can ignore estimation errors, e.g. the Markov blanket here simply corresponds to the variables with non-zero coefficient in a population OLS regression over all predictors. Lastly, by Lemma 1 we have that in

the population setting intervening on each predictor variable once is sufficient to produce the correct estimate. This yields a limit on the number of iterations for which A-ICP has to be run. Hence, in the population setting we sample without replacement from the pool of possible intervention targets. This is also why the *empty-set* strategy is not applicable in the population setting—we never intervene on the same variable twice in any case. This stands in contrast to the finite regime.

We compare the performance of the two proposed policies ("markov" and "markov + r") with each other and a baseline *random* policy which picks intervention targets at random from all predictors. In Figure 2, we compare how quickly the policies recover the causal parents in terms of the Jaccard similarity between the estimate $\hat{S}$ and the truth $S^*$: $|\hat{S} \cap S^*|/|\hat{S} \cup S^*|$. This metric is equal to one if and only if $\hat{S} = S^*$. Furthermore, we assess how many interventions the policies require in total to achieve *exact* recovery. Note that when the Markov blanket is composed of just the parents, both proposed policies are equivalent. This is the case for 405 out of the 1000 SCMs. Therefore, we plot the results for the remaining graphs separately in Figure 2 (panels (b) and (d)). In both of the considered metrics, combining the *Markov* and *ratio* strategies produces the best performing policy.

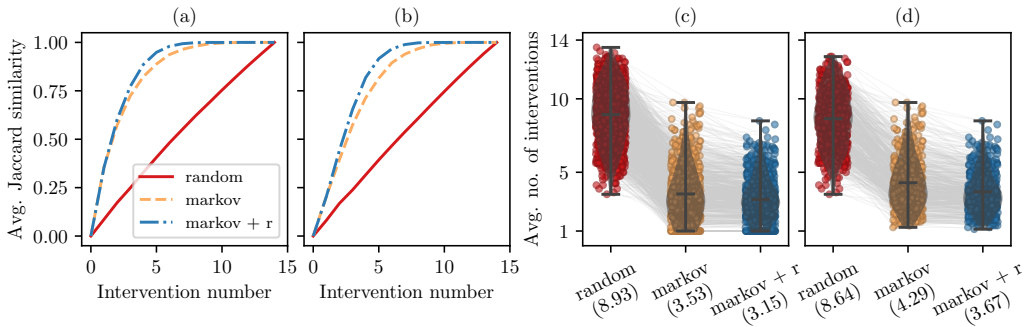

Figure 2: (population setting) Left: Average Jaccard similarity as a function of the number of interventions, for all 1000 SCMs of size 15 (a), and those for which the Markov blanket contains more than just the parents (b). In the first intervention the policies "markov" and "markov + r" perform equally, as they decide among the same pool of intervention targets (the Markov blanket). After interventions on all predictors all direct causes are determined. Right: Average number of interventions needed to achieve exact recovery of the causal parents, for each one of the 1000 SCMs (c) and those for which the Markov blanket contains more than just the parents (d). Each SCM is represented by a dot and connected across policies by a line. The total average of interventions employed by each policy is given below its label.

## 5.2 Finite sample setting

For finite samples, we individually evaluate the effect of the three strategies put forward in section 4, as well as combinations of them. In total we have 7 policies, each using a different combination of strategies, plus the random baseline policy. For the sample allocation, we fix the size of the sample collected per intervention; we perform experiments for 10, 100 and 1000 observations per sample. The same metrics as in section 5.1 are shown in Figures 3 and D.5[2]. As real experiments tend to be extremely costly in terms of time and money, the goal is to achieve good performance after as few interventions as possible. Hence, we here focus on the first 20 iterations of A-ICP while the total number of iterations is 50. Figures D.8 and D.7 show the performance over all 50 iterations.

The results show interesting patterns. In general, we observe that the choice of the optimal strategy depends on the number of observations one can allocate to each intervention and how many interventions can be performed in total. Relying on the Markov blanket estimate (policies labeled with "markov") leads to good initial but poor performance for larger $t$, independently of what other strategies are used. This estimate is obtained by performing an L1-regularized least squares regression on all predictors (i.e. the Lasso [36]), picking the regularization parameter for each SCM by cross

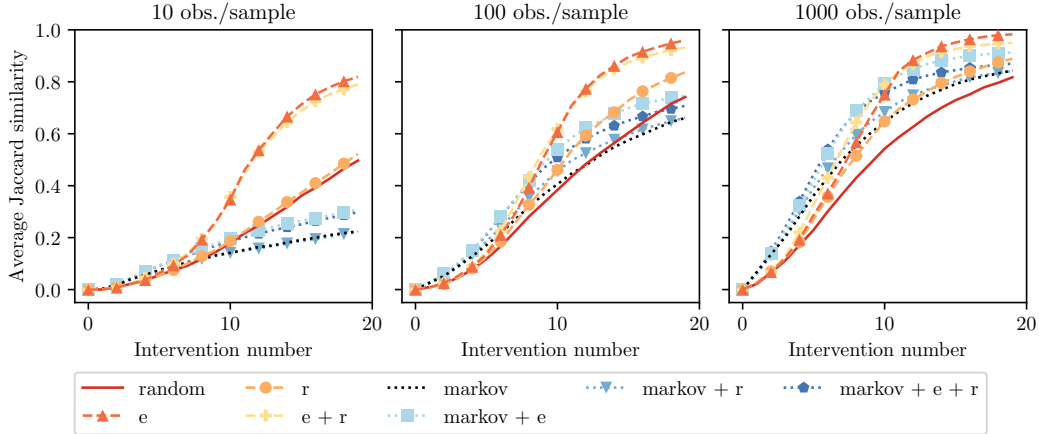

Figure 3: (finite regime) Average Jaccard similarity for 300 SCMs of size 12 as a function of the number of interventions for 10, 100 and 1000 observations per sample. Here, $\alpha = 0.01$.

validation. In the first few iterations, the policies quickly identify the parents contained in the estimate, as can be seen in Figure 3. However, when not all parents are contained in the estimate the policies become stuck performing non-informative interventions. As a result, for many of the SCMs not all direct causes are recovered after reaching the limit of iterations, clearly seen for 50 iterations in Figure D.7 (Appendix D). This problem is alleviated at larger sample sizes, where the Lasso yields a better estimate of the Markov blanket. We provide further error analyses in Appendix D.4.

In principle, the other two strategies may also suffer from estimation errors: For the *empty-set* strategy, the empty set may be wrongly accepted after an intervention on a parent, which is then discarded from future interventions. This problem of statistical power is attenuated for larger sample sizes and higher intervention strengths. For the *ratio* strategy, falsely rejecting stable sets and wrongly accepting unstable sets can bias the estimate of the stability ratio of some parents and thereby keep A-ICP from intervening on such parents. Hence, if there are no constraints on the number of total interventions, the *random* baseline policy is the most robust option as it is not as affected by estimation errors as the other policies. However, for small $t$, it is clearly outperformed by most of the other policies. As can be seen in Figure 3, the gain over the *random* policy becomes larger as the sample size increases.

For our experimental settings, we find that the performance of the *empty-set* strategy is quite robust, outperforming the remaining policies across the different sample sizes and a large range of intervention numbers. Using the *Markov* and/or the *ratio* strategy in addition only yields clear improvements for larger sample sizes. In Appendix D.5 we further analyze the performance for different intervention strengths and significance levels. Importantly, while the discussed possible estimation errors affect the choice of the optimal intervention target selection, the ICP error control on the estimate $\hat{S}$ is unaffected by this and remains intact (also see Figure D.6).

## 5.3 Comparison with ABCD

We compare the performance of A-ICP against that of the Bayesian ABCD strategy [1]. We choose this strategy as it allows directly learning the parents of the response. It hence lends itself to a more fair comparison than strategies which estimate the full graph. That said, the comparison is still not straightforward, as both strategies make different assumptions. ABCD requires a large observational sample, which it then uses to sample from the posterior through a bootstrap procedure based on GIES [10], but the interventional sample size can be as small as one. A-ICP does not rely on a large observational sample but is regression-based and requires more than one observation per intervention. We establish a middle ground by providing both methods with an observational sample of size 1000 and 10 observations per intervention. In Figure 4, we compare the methods over 50 iterations in terms of (1) the Jaccard similarity, and (2) the family-wise error rate (FWER) $\hat{P}(\hat{S} \nsubseteq S^*)$, i.e. the probability of having one or more false positives in the estimate of the causal parents. The results for

varying observational sample sizes are similar and can be found in Appendix D.3. Details about the experimental setup are given in Appendix E.

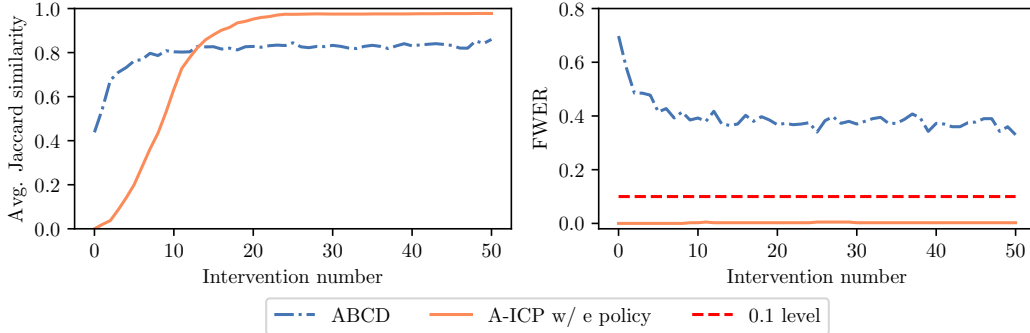

Figure 4: Average Jaccard similarity and family-wise error rate of the ABCD and A-ICP estimates, as a function of the number of interventions. Exploiting high probability candidates from the posterior over graphs based on the observational sample, ABCD shows better results in terms of Jaccard similarity in the first few iterations while not approaching a Jaccard similarity of one for large $t$. The good initial performance comes at the cost of false positives which are only reduced somewhat as $t$ increases. In contrast, A-ICP remains conservative in the first few iterations, often returning the empty set as an estimate. While A-ICP retains its error control over all iterations, its power increases steadily with the number of interventions, approaching a Jaccard similarity of one. Here, $\alpha = 0.1$.

## 6  Discussion

In section 2 we characterize the effect of interventions on the stability of sets of predictors. We leverage these results to construct several intervention selection policies for A-ICP. We find that the *empty-set* strategy shows good performance across different sample sizes and number of performed interventions. The *ratio* and the *Markov* strategy additionally yield improvements for larger sample sizes and for small intervention numbers. All policies outperform the *random* baseline policy across a large range of settings. While ICP is often criticized for its lack of power, we see that A-ICP can quickly overcome this weakness while maintaining the guarantees of ICP.

We welcome the discussion whether the proposed policies for A-ICP could be improved as many interesting questions remain. ICP does not require knowledge of the intervention locations in each environment. This makes it robust to interventions with off-target effects, i.e. effects on variables other than the target. Furthermore, A-ICP allows for combining data from existing environments with possibly unknown intervention targets with data from experiments that are performed with this knowledge. On the other hand, one might ask whether, since we know the intervention location when running A-ICP, we discard useful information. Of the proposed policies, only the ones that use the *empty-set* strategy leverage this information.

Finally, the results from section 2 are quite general in the sense that they make no assumptions on the function class or noise distributions of the SCM. As such, it would be interesting to assess to what extent A-ICP improves the power of more general extensions of Invariant Causal Prediction, such as nonlinear ICP [14] or ICP for sequential data [29]. Overall, this work shows that in an active learning setting, one can construct competitive methods with invariance as the underlying principle for causal discovery.

## Discussion of broader impact

Any method that learns from finite data is subject to statistical estimation errors and model assumptions that necessarily limit the full applicability of its findings. Unfortunately, study outcomes are not always communicated with the required qualifications. As an example, statistical hypothesis testing is often employed carelessly, e.g. by using p-values to claim "statistical significance" without paying attention to the underlying assumptions [5]. There is a danger that this problem gets exacerbated when one aims to estimate causal structures. Estimates from causal inference algorithms could be claimed to "prove" a given causal relationship, ruling out various alternative explanations that one would consider when explaining a statistical association. For example, ethnicity could be claimed to have a causal effect on criminality and thereby used as a justification for oppressive political measures. While this would represent a clear abuse of the technology, we as researchers have to ensure that similar mistakes in interpretation are not made unintentionally. This implies being conscientious about understanding as well as stating the limitations of our research.

While there is a risk that causal inference methods are misused as described above, there is of course also an array of settings where causal learning—and in particular active causal learning—can be extremely useful. As our main motivation we envision the empirical sciences where interventions correspond to physical experiments which can be extremely costly in terms of time and/or money. For complex systems, as for example gene regulatory networks in biology, it might be difficult for human scientists to choose informative experiments, particularly if they are forced to rely on data alone. Our goal is to develop methods to aid scientists to better understand their data and perform more effective experiments, resulting in significant resource savings. The specific impact of our proposed methodology will depend on the application. For the method we propose in this work, one requirement for application would be that the experiments yield more than one data point (and ideally many), so that our invariance-based approach can be employed. In future work, we aim to develop methodology that is geared towards the setting where only very few data points per experiment are available.

## Acknowledgments and Disclosure of Funding

We would like to thank Niklas Pfister, Jonas Peters, Armeen Taeb, Brian McWilliams and Nicolai Meinshausen for valuable discussions and comments on the manuscript. The research leading to these results was supported by a grant from the "la Caixa" Foundation (ID 100010434), with code LCF/BQ/EU18/11650051.

## Footnotes

[1]While Figure 1 is illustrated using do-intervention notation, noise or shift interventions are also possible, on a single or multiple variables.

[2]Additionally, we plot the family-wise error rate $\hat{P}(\hat{S} \not\subseteq S^*)$ in Figure D.6 (Appendix D.2.1), confirming that the ICP error control is indeed maintained.

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
