[Supplementary Material]

# A Intervention stable sets, plausible causal predictors and informative interventions

## A.1 Intervention stable sets

A set of predictors $S$ is an intervention stable set if it d-separates the response from all interventions, i.e. if the d-separation statement $I \perp\!\!\!\perp_{\mathcal{G}} Y | X_S$ holds in $\mathcal{G}(\mathcal{S}^{\mathcal{E}})$ for all interventions $I$ active in $\mathcal{E}$. An example follows:

**Example A.1.** Let $\mathcal{E}$ be a collection of environments with direct interventions on $X_0$ and $X_4$, as shown in the graph. Then, the intervention stable sets are

$$\mathbb{S}_{\mathcal{E}} = \{0\},$$
$$\{0, 4\},$$
$$\{0, 3, 4\}$$
$$\{0, 1\}$$
$$\{0, 1, 4\}$$
$$\{0, 1, 3, 4\}.$$

## A.2 Stable sets vs. plausible causal predictors

While $\mathbb{S}_{\mathcal{E}} \subseteq \mathbb{C}_{\mathcal{E}}$, it is not generally true that $\mathbb{S}_{\mathcal{E}} = \mathbb{C}_{\mathcal{E}}$. Importantly, this does not change when assuming faithfulness as the following example illustrates.

**Example A.2.** Take the following SCM,

$$X_0 := \varepsilon_0$$
$$X_1 := w_{01} X_0 + \varepsilon_1$$
$$X_2 := w_{02} X_0 + w_{12} X_1 + \varepsilon_2$$

with $\varepsilon_i \sim_{\text{i.i.d.}} \mathcal{N}(\mu_i, \sigma_i^2)$ noise variables such that $\varepsilon_i \perp\!\!\!\perp \varepsilon_j \ \forall i, j$. Consider $Y := X_1$ and the conditioning sets $S_0 = \{0\}$ and $S_2 = \{2\}$. In the following, we assess the invariance of the conditional distributions $Y|X_0$ and $Y|X_2$ under interventions. The conditional distributions of $Y|X_0$ and $Y|X_2$ are both Gaussian and below we compute their expectations and variances. For $Y|X_0$ we have:

$$\mathbb{E}(Y|X_0) = \mathbb{E}(Y) + \frac{\text{Cov}(Y, X_0)}{\text{Var}(X_0)}(X_0 - \mathbb{E}(X_0))$$

$$= w_{01}\mu_0 + \mu_1 + \frac{w_{01}\sigma_0^2}{\sigma_0^2}(X_0 - \mu_0) = \mu_1 + w_{01}X_0$$

$$\text{Var}(Y|X_0) = \text{Var}(Y) - \frac{\text{Cov}(Y, X_0)^2}{\text{Var}(X_0)} = w_{01}^2\sigma_0^2 + \sigma_1^2 - \frac{(w_{01}\sigma_0^2)^2}{\sigma_0^2} = \sigma_1^2$$

For $Y|X_2$ we have:

$$\mathbb{E}(Y|X_2) = \mathbb{E}(Y) + \frac{\text{Cov}(Y, X_2)}{\text{Var}(X_2)}(X_2 - \mathbb{E}(X_2))$$

$$= w_{01}\mu_0 + \mu_1 + \frac{\sigma_0^2(w_{01}w_{02} + w_{01}^2 w_{12}) + w_{12}\sigma_1^2}{\sigma_0^2(w_{02}^2 + w_{12}^2 w_{01}^2 + 2w_{02}w_{12}w_{01}) + w_{12}^2\sigma_1^2 + \sigma_2^2}(X_2 - \mathbb{E}(X_2))$$

$$\mathrm{Var}(Y|X_2) = \mathrm{Var}(Y) - \frac{\mathrm{Cov}(Y, X_2)^2}{\mathrm{Var}(X_2)}$$

$$= w_{01}^2\sigma_0^2 + \sigma_1^2 - \frac{(\sigma_0^2(w_{01}w_{02} + w_{01}^2 w_{12}) + w_{12}\sigma_1^2)^2}{\sigma_0^2(w_{02}^2 + w_{12}^2 w_{01}^2 + 2w_{02}w_{12}w_{01}) + w_{12}^2\sigma_1^2 + \sigma_2^2}$$

If we additionally assume $\mu_i = 0$, $w_{ij} = 1 \; \forall i, j$ and $\sigma_1^2 = \sigma_2^2 = 1$, the above expressions become

$$\mathbb{E}(Y|X_2) = \frac{1}{2}X_2 \qquad \text{and} \qquad \mathrm{Var}(Y|X_2) = \sigma_0^2 + 1 - \frac{(2\sigma_0^2 + 1)^2}{4\sigma_0^2 + 2} = \frac{1}{2}.$$

Consider now an intervention on $X_0$. We have that $S_0 = \{0\}$ is intervention stable and a set of plausible causal predictors. On the other hand, $S_2 = \{2\}$ does not d-separate $Y$ from the intervention on $X_0$, and is not intervention stable; however, for interventions that affect only the variance of $X_0$ (i.e. $\sigma_0^2$), $S_2$ is a set of plausible causal predictors. Under this setting, we have that $\mathbb{S}_{\mathcal{E}} \subset \mathbb{C}_{\mathcal{E}}$.

Example A.2 shows that $\mathbb{S}_{\mathcal{E}} \neq \mathbb{C}_{\mathcal{E}}$. However, one might ask how often this happens in practice. In the example, this only happens when we set the weights, means and variances to very particular values. When these parameters are sampled from a continuous distribution, we conjecture that the set of parameters for which $\mathbb{S}_{\mathcal{E}} \neq \mathbb{C}_{\mathcal{E}}$ has probability zero. We call the assumption that $\mathbb{S}_{\mathcal{E}} = \mathbb{C}_{\mathcal{E}}$ *stability-faithfulness*.

### A.3 Informative interventions

If we make the assumption that $\mathbb{C}_{\mathcal{E}} = \mathbb{S}_{\mathcal{E}}$, by Lemma 1 we know that, in the absence of estimation errors, a direct intervention on a parent is sufficient for it to appear in the ICP estimate. However, it is not a necessary condition, as is shown in the following example.

**Example A.3.** Let $\mathcal{E}$ be a collection of two environments: one without interventions and one with a direct intervention on $X_2$, as shown in the graph. The intervention stable sets are

$$\mathbb{S}_{\mathcal{E}} = \{0, 1, 2\},$$
$$\{0, 1, 3\},$$
$$\{0, 1, 2, 3\}.$$

Therefore,

$$S(\mathcal{E}) = \bigcap_{S : S \in \mathbb{S}_{\mathcal{E}}} S = \{0, 1\},$$

which shows that parents can appear in the intersection of intervention stable sets without being directly intervened on. In this case, a direct intervention on $X_2$ is very informative, as it reveals two parents simultaneously. To the best of our knowledge it is not clear when situations like the above arise, or how they can be detected from the accepted sets. Therefore, as a first approach we consider direct interventions on the parents as "maximally informative", and the goal of the proposed policies is to pick such interventions.

## B Detailed description of ICP

Here we present a slightly adapted version of Invariant Causal Prediction [27]. In contrast to the original formulation, algorithm 2 takes `candidate sets` as an additional, optional argument. If `candidate sets` is not provided, algorithm 2 corresponds to the original ICP formulation where the null hypothesis $H_{0,S}$ needs to be tested for all subsets of the predictors. As detailed in Corollary 3.1, A-ICP (algorithm 1) does not require testing all subsets in each iteration. Hence, when ICP is called as a subroutine in A-ICP only the `accepted sets` from the previous iteration are provided as `candidate sets` to ICP.

In general terms, the null hypothesis $H_{0,S}$ states that the distribution of the response $Y$ conditional on the predictors $X_S$ is invariant across the different environments. Depending on which ICP version is employed, the specific formulation of null hypothesis is adapted to the respective problem setting. In the linear case, one can test for the equality of the regression coefficients and the noise variances across environments but other options are also possible (for details, please see [27]). When using nonlinear ICP [14], the environment is considered as an additional variable $E$ in the system and the null hypothesis then corresponds to $Y \perp\!\!\!\perp E | X_S$ which is tested using a non-parametric conditional independence test. To formulate the algorithm below generically, we leave open what formulation and test is chosen for $H_{0,S}$.

---

**Algorithm 2:** ICP

---

**Output :** `accepted sets` sets for which the null hypothesis cannot be rejected,
      $\hat{S}(\mathcal{E})$ estimate of the parents of the response
**Input  :** i.i.d. samples of $(X, Y)$ from different environments $\mathcal{E}$,
      `candidate sets` sets for which to test the null hypothesis,
      $\alpha$ significance level
**if** `candidate sets` is `null` **then**
  | `candidate sets` $\leftarrow S \subseteq \{1, \ldots, p\}$
**end**
**for each** $S$ *in* `candidate sets` **do**
  | Test whether $H_{0,S}$ holds at level $\alpha$.
**end**
$\hat{S}(\mathcal{E}) := \bigcap_{S: H_{0,S} \text{ not rejected}} S$;
`accepted sets` $\leftarrow \{S \mid H_{0,S} \text{ not rejected}\}$;
**return** `accepted sets`, $\hat{S}(\mathcal{E})$

---

## C   Analysis of computational complexity

The runtime of A-ICP depends on (i) the runtime of ICP, and (ii) the runtime of the chosen intervention selection policy. The runtime of ICP depends on the complexity $c(N, e, k)$ of testing the invariance from Eq. (1) for a set of predictors $S$ of size $k$ over a total of $N$ observations from $e$ environments. Thus, for each iteration of A-ICP, the cost of running ICP on all sets of predictors is

$$\sum_{S \subseteq \{1, \ldots, p\}} c(N, e, |S|) \leq 2^p c(N, e, p).$$

Furthermore, let $s(N, e, p)$ denote the complexity of the chosen intervention selection policy. In this notation, the complexity of running A-ICP for a total of $T$ iterations is

$$O(T 2^p c(N, e, p) s(N, e, p)). \tag{2}$$

In the experiments of section 5, we test invariance by performing a least-squares regression of the response on the predictors, and then running a two-sample t-test and an F-test [27, section 3.1.2] over the residuals. Under this approach, the complexity of testing a single set of predictors of size $k$ is the cost of performing a least-squares regression and computing the residuals ($O(k^2 N)$) and the cost of performing the t-test and F-test over each split of the $e$ environments ($O(eN)$). Thus, $c(N, e, k) = O(N(k^2 + e))$.

The cost of the empty-set strategy corresponds to that of testing the empty set over the initial and current environments, i.e. $c(N, 2, 0)$. For the ratio strategy, one must compute the stability ratio (c.f. Definition 2.2), which in the worst case (all sets are accepted) incurs a cost of $O(p 2^p)$. The Markov strategy carries the cost of performing a Lasso regression in the first iteration, which is dominated by the other terms in Equation 2.

By Corollary 3.1, at each iteration it suffices for ICP to consider only the sets accepted in the previous iteration. On average, this provides a substantial speed up as not all $2^p$ subsets of predictors need to be re-tested in each iteration. However, the complexity is still exponential in the number of variables $p$, which limits the applicability of A-ICP to "large $p$" settings. Nonetheless, A-ICP can still be useful in settings where the time needed to carry out an experiment far outweighs the computation time to select the next experiment, which is common in empirical sciences like biology.

# D    Additional experimental results

Here, we present additional experimental results. In section D.1, we show the average number of interventions until exact recovery (Figure D.5) for the finite-sample experiments presented in section 5. In section D.2, we provide additional results for the total 50 iterations over which the policies are run: the family-wise error rate is shown in Figure D.6, Figures D.8 and D.7 show the Jaccard similarity and the average number of iterations until exact recovery, respectively. The error analysis of the Markov blanket estimation procedure is displayed in section D.4, Figure D.10. In section D.3, we present the results from running ABCD and A-ICP with different sizes of the initial observational sample. Finally, section D.5 contains additional results comparing the interplay between the A-ICP significance level and the performance for different intervention strengths.

## D.1    Average number of interventions for exact recovery

Figure D.5: (finite regime) Average number of interventions until the causal parents are recovered exactly for the first 20 iterations of A-ICP, for each one of the 300 SCMs. If $\hat{S} \neq S^*$ at $t = 20$, we set the statistic to 20. For each SCM, we average the performance over the 8 different random seeds considered. The average performance of each SCM is represented by a dot and connected across policies by a grey line. The total average of interventions employed by each policy is given below its label. The "e" policy performs well across all sample sizes, and is the best performer except at 1000 obs./sample where it falls behind the "e + r" and "Markov + e" policies.

## D.2 Results for 50 interventions

We run the policies for a total of 50 interventions, to evaluate their performance in a setting where more experimental rounds are possible.

### D.2.1 Family-wise error rate

In Figure D.6 we plot the family-wise error rate (FWER) $\hat{P}(\hat{S} \nsubseteq S^*)$. Recall that to achieve FWER control across all iterations, we have to apply a correction to the level at which ICP is run in each iteration of A-ICP (also see Appendix B, algorithm 1). Due to the strong dependence between the tests, we use a Bonferroni correction by running ICP at iteration $t$ at the level $\alpha/T$ where $\alpha$ is the overall significance level and $T$ is the total number of iterations. Figure D.6 confirms that the FWER is indeed kept below the $0.01$ significance level at which A-ICP is run, maintaining the coverage guarantees provided by Invariant Causal Prediction (ICP). The FWER lies well below the nominal level of $0.01$ due to the construction of the estimate $\hat{S}$. The error control rests on the fact that the true set of causal parents is rejected with probability smaller than $\alpha/T$ in each round of A-ICP. However, even if a mistake is made and the true set is rejected, accepting other sets and computing their intersection to obtain $\hat{S}$ may still result in $\hat{S} \subseteq S^*$.

Figure D.6: (finite regime) Family-wise error rate (FWER) for the finite-sample experiments. The FWER $\hat{P}(\hat{S} \nsubseteq S^*)$, i.e. the probability of wrongly marking as direct causes variables which are not, is kept below the $0.01$ significance level at which A-ICP is run, maintaining the coverage guarantees provided by Invariant Causal Prediction (ICP) [27].

### D.2.2 Jaccard similarity and average number of interventions for exact recovery

The results in Figure D.7 and Figure D.8 illustrate the fact that if there are no constraints on the number of interventions, the *random* policy is among the most robust options, as its choice of intervention targets is unaffected by estimation errors. However, it needs a large number of iterations to achieve competitive performance and only achieves an average Jaccard similarity close to one when $t$ approaches 50.

Overall, the *empty-set* strategy is the best performer across all sample sizes for a large range of intervention numbers. For the *Markov* policies, the issues arising from obtaining an estimate of the Markov blanket are more apparent in this setting: while the policies quickly identify parents contained in the estimate, they become stuck performing non-informative interventions and fail to identify the remaining parents for some SCMs. This can be seen in Figure D.7 which shows the average number of interventions needed to achieve exact recovery (averaged over different random seeds).

While at 1000 obs./sample, combining the *ratio* strategy with the *empty-set* strategy grants an advantage in performance over using the *empty-set* strategy alone for the early iterations, this advantage is lost later on as the combination performs worse for some particular graphs, which

decreases the average performance. Combining the *ratio* with the *empty-set* strategy can in some cases be less effective than the *empty-set* strategy alone for the following reasons. First, the interventions chosen here are quite strong such that the *empty-set* strategy is *not* affected by the issue of statistical power that the empty set may be wrongly accepted after an intervention on a parent, which would then be discarded from future interventions. Second, in the finite regime, it is not necessarily sufficient to intervene on a parent once for it to appear in $\hat{S}$ due to a lack of power. In other words, after an intervention on a parent not all unstable sets are necessarily rejected. In contrast, intervening on children of the response can sometimes lead to a larger number of unstable sets being rejected and hence an estimate $\hat{S}$ with larger Jaccard similarity. Intervening on children of the response tends to occur more often when using the *empty-set* strategy alone. Lastly, the *ratio* strategy is subject to the following testing errors: falsely rejecting stable sets and wrongly accepting unstable sets can bias the estimate of the stability ratio of some parents and thereby keep A-ICP from intervening on such parents. Since a rejected set is not re-tested at future iterations (by Corollary 3.1), falsely rejecting a stable set at some iteration $t$ will also bias the estimate of the stability ratio for future iterations (as long as the set remains stable). While this discussion highlights the failure cases of the *ratio* strategy, the analysis in section D.5 shows that for smaller intervention strengths the *empty-set* strategy is not always the best-performing policy, presumably due to the power issue described above.

Figure D.7: (finite regime) Average number of interventions until the causal parents are recovered exactly for $T = 50$, for each one of the 300 SCMs. If $\hat{S} \neq S^*$ at $t = 50$, we set the statistic to 50. For each SCM, we average the performance over the 8 different random seeds considered. Each SCM is represented by a dot and connected across policies by a grey line. The total average of interventions employed by each policy is given below its label.

Figure D.8: (finite regime) Average Jaccard similarity for 300 SCMs of size 12 as a function of the number of interventions for 10, 100 and 1000 observations per sample. Here, $\alpha = 0.01$ and the policies' performance is shown for all 50 iterations.

## D.3 Effect of the observational sample size on the performance of ABCD and A-ICP

Figure D.9: Average Jaccard similarity (top) and family-wise error rate (FWER) (bottom) of the ABCD and A-ICP estimates, as a function of the number of interventions (50 in total). Both the Jaccard similarity and family-wise error rate of ABCD are affected by the size of the initial observational sample. While the performance of A-ICP improves slightly with a larger observational sample, having a large observational sample is not a requirement of A-ICP. In all cases, A-ICP maintains the FWER under the desired level ($\alpha = 0.1$) while its power increases steadily with the number of interventions. Details about the experimental setup can be found in Appendix E.

For the results summarized in Figure D.9, we vary the size of the initial observational sample while keeping the number of observations per interventional sample fixed at 10. For the observational sample we consider 50, 100 and 1000 observations. ABCD requires a large observational sample to obtain a sufficiently good estimate of the posterior over graphs. This leads to relatively large Jaccard similarities for the first few iterations. In contrast, A-ICP remains conservative at the beginning, often returning the empty set as an estimate as a large number of predictor sets are stable for small $t$. While A-ICP controls the nominal FWER of $\alpha = 0.1$ over all iterations, its power increases steadily with the number of interventions, reaching an average Jaccard similarity close to one for large $t$. In contrast, ABCD does not control the false positives: while the average Jaccard similarity increases with the number of iterations, it does not approach one since the estimate still contains false positives even for large $t$. The comparison for different observational sample sizes shows that both the average Jaccard similarity and the FWER improve for ABCD the larger the initial observational sample is.

## D.4 Error analysis of the Markov blanket estimation procedure

Figure D.10: Error analysis of the Markov blanket estimation procedure for the finite regime, for 1000 SCMs of size 12. The estimate is produced by taking the variables with non-zero coefficients resulting from a Lasso regression over all predictors in the sample from the observational environment. The regularization parameter is picked individually for each SCM by ten-fold cross-validation. By size we refer to the number of variables included in the estimate. As expected, the quality of the estimate improves with the sample size. However, even at the largest sample size, for some SCMs not all the parents are contained in the estimate. In these cases the policies relying on the *Markov* strategy become stuck performing non-informative interventions, and fail to recover all parents after the limit $T$ of iterations is reached.

In figure D.10 we provide further analyses to understand the behavior of the policies using the *Markov* strategy. For 1000 SCMs of size 12, the Markov blanket is estimated with the Lasso using the observational sample. The regularization parameter is chosen by ten-fold cross-validation. In Figure D.10 we plot (i) the proportion of estimates which contain the Markov blanket (top left); (ii) the average size of the estimate where size refers to the number of variables included in the estimate (top right); (iii) proportion of estimates which contain all parents (bottom left); and (iv) the average size of the estimate when it contains all parents (bottom right). At smaller sample sizes, often not all parents are included in the estimate (bottom left). Hence, policies using the *Markov* strategy do not intervene on them which often results in a failure to identify them. While this issue is attenuated for larger sample sizes, it does not disappear entirely, even for a sample size of 1000. This explains why the policies using the *Markov* strategy have a lower average Jaccard similarity for large $t$, as can be seen in Figure D.8.

## D.5 A-ICP significance level and intervention strength

Figure D.11: (finite regime) Performance of the policies for an observational sample of size 100 and 10 observations per interventional sample, for different intervention strengths (rows), and significance levels of A-ICP (columns). Overall, the gains in performance over the *random* policy increase with the intervention strength. The performance of the *empty-set* policy increases with the level, as power also increases and the empty set is rejected more often. The *ratio* policy is largely unaffected by the change in level, and often yields additional improvements when used in combination with the *empty-set* strategy in the initial iterations.

To correct for multiple testing of the accepted sets, we apply a Bonferroni correction to the significance level of the statistical tests performed in each round of A-ICP (see algorithm 1). To assess the

sensitivity of the results with respect to the overall significance level of A-ICP $\alpha$ and the intervention strength, we run A-ICP at 100 observational data points and 10 observations per interventional sample for $\alpha \in \{0.005, 0.01, 0.05, 0.1\}$, and shift interventions with variance 1 and means $3, 5$ and $7$. Details about the experimental settings can be found in Appendix E.

Figure D.11 shows that the *ratio* and the *empty-set* strategy yield larger improvements over the *random* policy for larger intervention strengths. This is to be expected as statistical power increases with the intervention strength and both the *ratio* and the *empty-set* strategy rely on statistical testing to choose the intervention target. While the results reported in section 5 are based on experiments with strong interventions (shift interventions with variance 1 and mean 10), the relative performance between the *ratio* and the *empty-set* strategy changes when considering weaker interventions. For instance, for interventions with mean 5 (second row), the *empty-set* strategy does not reach an average Jaccard similarity of one for $t = 50$. For large $t$, the *ratio* as well as the *random* strategy perform better.

# E   Experimental settings

Table E.1: Overview of the experimental parameters considered. Below, $n_e$ denotes the number of interventions per interventional sample. For the ABCD experiments, we additionally vary the size of initial observational sample. Unless indicated otherwise, all interventions are shift interventions with different means $\mu$ and variance $\sigma^2 = 1$.

|  | # SCMs | # seeds | $n_e$ | $p$ | $T$ | $\alpha$ | Interventions |
|---|---|---|---|---|---|---|---|
| Population | 1000 | 8 | — | 15 | 15 | — | $\mu = 10$ |
| Finite regime | 300 | 8 | $\{10, 100, 1000\}$ | 12 | 50 | .01 | $\mu = 10$ |
| ABCD | 100 | 4 | 10 | 12 | 50 | .1 | do, $\mu = 9$ |
| Figure D.11 | 100 | 4 | 10 | 12 | 50 | $\{.005, .01, .05, .1\}$ | $\mu \in \{3, 5, 7\}$ |

The code to reproduce the experimental results is provided in the repositories `https://github.com/juangamella/aicp` and `https://github.com/juangamella/abcd`. Additionally, to generate synthetic interventional data we make the python package `sempler` (`https://github.com/juangamella/sempler`) available.

**Population setting**   For the experiments, 1000 linear structural causal models of size 15 are randomly generated by sampling from Erdős-Rényi graphs with an average degree of 3. The weights are sampled uniformly at random from $[0.5, 1]$, and the intercepts and noise variances from $[0, 1]$. In the population setting no further assumptions are made on the noise distributions, besides having finite mean and variance to perform the OLS regression. To perform the regression in the population setting, we maintain a symbolic representation of distributions that contains their first and second moments, and allows conditioning and marginalization. Further experiments with SCMs of different size and parameters yielded very similar results to the ones presented in the main text and are not shown separately. For every SCM, each policy is run 8 times with different random seeds, to account for the stochastic component of the policies.

**Finite sample setting**   For the experiments, 300 linear structural causal models of size 12 are randomly generated, again by sampling from Erdős-Rényi graphs with an average degree of 3. The weights, variances and intercepts are sampled as in the population setting. Interventions are shift-interventions with mean 10 and variance 1. Like in the population setting, the policies are run 8 times with different random seeds, for 50 iterations. To simplify the implementation, we assume that the underlying noise distributions are Gaussian, and set ICP to use a two-sample t-test and F-test to check the invariance of the conditional distribution of the response. It is important to note that this is not a necessary requirement: the results derived in section 2 (e.g. Corollary 2.1) apply to arbitrary SCMs with arbitrary noise distributions, and ICP can use other statistical tests, including non-parametric ones. However, we expect that the effect of the sample size on the results will be different under different noise distributions and tests. Figure 3 corresponds to the results of running A-ICP at a significance level of $0.01$.

**Comparison with ABCD** We randomly generate 100 linear structural causal models of size 12, by sampling from Erdős-Rényi graphs with an average degree of 3. The weights, variances and intercepts are sampled as in the population setting. ABCD requires a Gaussian SCM, so the underlying noise distributions are Gaussian and ICP is set to use a two-sample t-test and F-test to check the invariance of the conditional distribution of the response. At each iteration, each method receives 10 observations from the newly performed intervention. Interventions are do-interventions, as this is the only type of intervention that the ABCD implementation considers. Experiments are carried out for different sizes of the initial observational sample (see Figure D.9), running each method a total of 4 times to account for stochasticity. The output of ABCD are posterior probabilities over parent sets; the average Jaccard similarity and FWER are computed by taking the argmax of the posterior. ABCD is set to use 100 bootstrap samples and A-ICP is run at a significance level of 0.1.

**Intervention strength vs. level (Figure D.11)** The experiments are run on 100 randomly generated linear structural causal models of size 12, sampled from Erdős-Rényi graphs with an average degree of 3. The remaining parameters are sampled as in the population setting. We then compare the performance of the *random*, *empty-set* and *ratio* strategies at different significance levels $(0.005, 0.01, 0.05$ and $0.1)$ and intervention strengths, i.e. we use shift interventions with variance 1 and means $3, 5$ and $7$. We collect 100 observations from the initial observational environment and 10 observations from each interventional environment. Again ICP employs a t-test and F-test to check the invariance of the conditional distribution of the response.

# F Proofs

To simplify notation, let $\mathrm{PA}(i)$ be the parents of $X_i$ and let $\mathrm{PA}(S) = \{j \in \{1, ..., p\} \mid \exists i \in S : j \in \mathrm{PA}(i)\}$ denote the parents of variables in a set $S$. Similarly, let $\mathrm{CH}(i)$ be the children of $X_i$ and let $\mathrm{CH}(S) = \{j \in \{1, ..., p\} \mid \exists i \in S : j \in \mathrm{CH}(i)\}$ denote the children of variables in a set $S$. Let $\mathrm{DE}(S) = \{j \in \{1, ..., p\} \mid \exists i \in S : j \in \mathrm{DE}(i)\}$ denote the descendants of variables in a set $S$. Note that the descendants of a variable include the variable itself, i.e. $i \in \mathrm{DE}(i)$.

**Lemma 1** (intervened parents appear on all intervention stable sets). *Let $\mathcal{E}$ be a set of observed environments and let $j \in PA(Y)$ be directly intervened on in $\mathcal{E}$. Then,*

$$S \subseteq \{1, ..., p\} \text{ is intervention stable} \implies j \in S.$$

*Proof.* Assume $S \subseteq \{1, ..., p\}$ is an intervention stable set such that $j \notin S$, and let $I$ denote the direct intervention on $j$. Then, there is a path $I \rightarrow j \rightarrow Y$ that is unblocked by $S$, which contradicts $S$ being intervention stable. $\square$

**Lemma 2** (sets containing descendants of directly intervened children are unstable). *Let $i \in CH(Y)$ be directly intervened on in $\mathcal{E}$. Then, any set $S \subseteq \{1, ..., p\}$ which contains descendants of $i$ is not intervention stable.*

*Proof.* Let $I$ denote the direct intervention on $i$, and let $S \subseteq \{1, ..., p\} : S \cap \mathrm{DE}(i) \neq \emptyset$. Then, the path $Y \rightarrow i \leftarrow I$ is not blocked by $S$. $\square$

**Lemma 3** (stability of the empty set). *Let $\mathcal{E}$ be any set of environments. Then,*

$$\emptyset \in \mathbb{S}_{\mathcal{E}} \iff \mathcal{E} \text{ contains no interventions on variables in } AN(Y).$$

*Proof.* ($\implies$) Assume the empty set is stable under environments $\mathcal{E}$ which contain an intervention $I$ on $j \in \mathrm{AN}(Y)$. Then there exists a path $Y \leftarrow ... \leftarrow j \leftarrow I$ which is not blocked by the empty set, arriving at a contradiction. ($\impliedby$) For every intervention $I$ on a variable $i$, every path from $Y$ to $I$ either

  (i) contains a collider, and is thus blocked by $\emptyset$, or

  (ii) does not contain a collider and is active under $\emptyset$.

Since $I$ is a source node, paths of type (ii) can only be of the form $Y \leftarrow ... \leftarrow i \leftarrow I$, which is not possible as $i$ would then be an ancestor of $Y$. $\square$

**Proposition 1** (ancestors appear on at least half of all stable sets). *Let $\mathcal{E}$ be any set of observed environments. Then, for any $j \in \{1, ..., p\}$,*

$$r_{\mathcal{E}}(j) < 1/2 \implies j \notin AN(Y).$$

*Proof.* We will prove the equivalent statement $j \in AN(Y) \implies r_{\mathcal{E}}(j) \geq 1/2$. For any $i \in \{1, ..., p\}$ we have that

$$r_{\mathcal{E}}(i) = \frac{|\{S \in \mathbb{S}_{\mathcal{E}} : i \in S\}|}{|\{S \in \mathbb{S}_{\mathcal{E}} : i \in S\}| + |\{S \in \mathbb{S}_{\mathcal{E}} : i \notin S\}|},$$

and therefore

$$r_{\mathcal{E}}(i) \geq 1/2 \iff |\{S \in \mathbb{S}_{\mathcal{E}} : i \in S\}| \geq |\{S \in \mathbb{S}_{\mathcal{E}} : i \notin S\}|. \tag{3}$$

We will show that for any $j \in AN(Y)$, and any intervention stable set $S$ such that $j \notin S$, the set $S \cup \{j\}$ is also intervention stable, satisfying the right hand side of Equation 3. To do this, we will use the fact that $S$ d-separates the response from all interventions, and show that the same is true for $S \cup \{j\}$, making it intervention stable.

Let $I$ denote an intervention on a variable $i$. For every path connecting $Y$ and the intervention, either

   (i) $j$ appears in the path as a collider,

  (ii) $j$ appears in the path but not as a collider,

 (iii) $j$ does not appear in the path but is downstream of a collider, or

 (iv) $j$ does not appear in the path and is not downstream of a collider.

If $S$ blocks paths of type (ii) and (iv), $S \cup \{j\}$ also does. Assume now there is a path of type (i) or (iii) which is blocked under $S$ but active under $S \cup \{j\}$. This implies that such path is blocked by a collider $c$ such that $j \in DE(c)$ and $S \cap DE(c) = \emptyset$; thus, there exists a path $Y \leftarrow ... \leftarrow j \leftarrow ... \leftarrow c \leftarrow ... i \leftarrow I$ which is active under $S$, i.e. $S \notin \mathbb{S}_{\mathcal{E}}$.

Therefore, for all $S \in \mathbb{S}_{\mathcal{E}}$ such that $j \notin S$, we have that $S \cup \{j\} \in \mathbb{S}_{\mathcal{E}}$, and

$$|\{S \in \mathbb{S}_{\mathcal{E}} : j \in S\}| \geq |\{S \in \mathbb{S}_{\mathcal{E}} : j \notin S\}| \implies r_{\mathcal{E}}(j) \geq 1/2.$$

$\square$

**Proposition 2** (intervention stable sets are plausible causal predictors). *Let $\mathcal{E}$ be a set of observed environments. Then, for all intervention stable sets $S \subseteq \{1, ..., p\}$, it holds that $S \in \mathbb{C}_{\mathcal{E}}$.*

*Proof.* The following is based on proof of proposition 3 in [30].

Let $\mathcal{E}$ be a set of observed environments, and let $S \in \mathbb{S}_{\mathcal{E}}$ be an intervention stable set. From [27] we know that $S$ is a set of plausible causal predictors iff $Y^e | X_S^e$ remains invariant for all environments $e \in \mathcal{E}$. Starting from setting 1, introduce an auxiliary random variable $E$ taking values in $\mathcal{E}$ with equal probability (for simplicity). To model the environments we construct an extended SCM $\mathcal{S}_{\text{full}}^{\mathcal{E}}$, where the variable $E$ appears on the assignments of the intervention variables $I$, and the assignments of the remaining variables remain as in $\mathcal{S}^{\mathcal{E}}$. As such, in $\mathcal{G}(\mathcal{S}_{\text{full}}^{\mathcal{E}})$ $E$ is a source node with only edges into the variables in $I$. The SCM $\mathcal{S}_{\text{full}}^{\mathcal{E}}$ induces a distribution $P_{\text{full}}$ over $(E, I, X, Y)$, which under setting 1 has a density $p$ that factorizes with respect to a product measure. Furthermore, since $P_{\text{full}}$ satisfies the Markov properties [24] and $S$ d-separates the response from all the intervention variables in $I$, it holds that $E \perp\!\!\!\perp Y \mid X_S \sim P_{\text{full}}$. Therefore, for every environment $e \in \mathcal{E}$, we have that

$$
\begin{aligned}
p(Y^e = y \mid X_S^e = x) &= p(Y = y \mid X_S = x, E = e) \\
&= \frac{p(Y = y \mid X_S = x)p(E = e \mid X_S = x)}{p(E = e \mid X_S = x)} \\
&= p(Y = y \mid X_S = x),
\end{aligned}
$$

and $Y^e \mid X_S^e$ remains invariant for all environments $e \in \mathcal{E}$.

$\square$