[Reviews · NeurIPS 2020]

Review 1

Summary and Contributions: This paper is about extending invariant causal prediction (ICP) to the active online setting. ICP is based on what has been described as the Most Useful Tautology Ever (MUTE): If we do not intervene on Y, then we do not intervene on Y. In other words, by not intervening on the response variable Y, then every other intervention is conditionally independent of Y given its parents. The authors exploit this fact along with certain properties of intervention stable sets to contribute 3 strategies for finding maximally informative interventions for discovering the parents of the response variable Y. These strategies consider both the population setting, where we can assume infinite data, and in the finite setting, where testing errors can occur. The authors claim these strategies can be adapted for both linear and nonparametric settings; and for single interventions, as well as batch interventions. The authors then perform experiments that demonstrate several different policy implementations of their strategies on simulated datasets from randomly chosen linear SCMs using single interventions, where the number of samples from each dataset ranges from the finite setting to the population setting. In the population setting, they compare two of their policies against a random intervention policy and in the finite setting, the compare seven policies against a random intervention policy. They also compare the previously published Active Budgeted Causal Design Strategy (ABCD) against one of their policies.

Strengths: Theoretical grounding: The authors provide well-motivated examples and counterexamples, proofs, algorithms of ICP and A-ICP, and well-documented python code implementations, not only of A-ICP, but also ICP and ABCD in the appendices. Empirical evaluation: The results the authors report indicate robust performance improvements in comparison to random strategies. They successfully demonstrate their policies are capable of finding the direct causes of the response variable with fewer interventions than a random policy. Compared to the ABCD strategy, they are more likely to find the full set of direct causes of the response variable. The relevance to the NeurIPS community is greatly enhanced by providing python notebooks that enable the reproduction of the figures and results.

Weaknesses: Theoretical grounding: The main manuscript is light on details about the algorithms due to space limitations, but the appendices more than make up for it. Empirical evaluation: The authors argue that their goal is to achieve good performance after as few interventions as possible. The ABCD strategy finds more direct causes with fewer interventions than any of the A-ICP policies. If I had a limited budget, I would probably use ABCD.

Correctness: The claims and methods appear to be correct. The reported results were a bit distorted, due to space limitations. Evaluating the empirical methodology with the appendices seems thorough.

Clarity: The paper assumes a lot of prior knowledge on the part of the reader. Fortunately, the authors do provide enough breadcrumbs to follow the citations to fill in missing gaps. For this paper, clarity is somewhat impeded due to space limitations, as there are a lot of results to report and not much space to report them. As a consequences, many of the most interesting contributions are buried in the appendices, as described above.

Relation to Prior Work: This paper stands on the theoretical results of two previously published papers: ICP and "Stabilizing Variable Selection and Regression" and more fundamentally, on MUTE. This distinguishes it from prior work in the active learning field that rely on bayesian or graphical approaches. The most similar work in terms of output type is ABCD, but they differ quite a bit in their input assumptions, which the authors try to resolve in order to make fair comparisons.

Reproducibility: Yes

Additional Feedback:


Review 2

Summary and Contributions: The authors proposed A-ICP, an active causal learning framework based on ICP [27] which assumes that the conditional distribution of the response, given its direct causes, remains invariant when intervening on arbitrary variables in the system. They proposed several active learning policies for performing interventions and used the results of intervetions to identify the parent set of the target variable.

Strengths: - Comparing with previous work in experiment design, the proposed approach does not need to know the parameters of the model (like in Bayesian approach) or knowing MEC (like in graph-theoretic approach). - In Proposition 1, a necessary condition for being an ancestor of the target variable is provided which is an interesting result.

Weaknesses: ===After rebuttal=== I read the reubttal and I think the proposed method has computational complexity issues and it should be compared with the naive solution of inteverening on the target variable (estimating MEC from finite sample size). Thus, I decided to keep my score unchanged. ================ - The main assumption of ICP may not be satisified in real world scenarios. In particular, in the linear model, it means that the variance of exognous noise of target variable should not be changed across environments. - It is not clear for me why we cannot intervene on the target variable and get enough samples to recover its parents. It might be a good idea to compare this solution with the proposed policies. - It is required to analyze the time complexity of the proposed policies mentioned in Section 4. - It is not clear whether each experiment consists of a single intervention or not in the proposed policies. It is better to clarify this issue.

Correctness: It seems that the claims and proposed methods are correct. However, I did not go into details of the proofs.

Clarity: The paper is generally well written.

Relation to Prior Work: The authors compared the proposed framework with other approaches in Introduction section.

Reproducibility: Yes

Additional Feedback:


Review 3

Summary and Contributions: The paper proposes an algorithm for active learning in causal models. In particular, they rely on Invariant Causal Prediction (ICP) to select which experiments should be performed. They further characterize causal effects on stable sets and then propose intervention selection policies.

Strengths: The paper considers a very important problem in causal inference: that is active learning on causal graphs. In particular, the goal is to construct efficiently the intervention stable sets as plausible causal predictors. the key tool is invariance: while the full causal graph may be non-identifiable, the conditional distribution of the response, given its direct causes, remains invariant. The proposed algorithm shows performance gains in empirical studies. The empty set policy outperforms the others across the different sample sizes and a large range of intervention numbers.

Weaknesses: The relationship between the sample size of the intervention experiments and the size of the causal graph. How much active learning can help with the analysis will depend on the sample size obtained in the intervention experiments relative to the size of the relevant variables (or loosely the signal to noise ratio in the analysis). While some experiments are performed in finite sample settings, it remains unclear how the "signal-to-noise" ratio might affect the relative performance of the algorithm. Further, empirical studies are limited mostly to linear settings. However, there exists some development in invariant causal prediction in nonlinear settings. These days causal inference does routinely work with nonlinear settings. How the conclusion drawn from linear settings can be generalized more generally would be an important and interesting discussion in the paper.

Correctness: The paper seems correct.

Clarity: The paper is quite well-written.

Relation to Prior Work: The prior is adequately discussed.

Reproducibility: Yes

Additional Feedback: See above. -------------- Thank you to the authors for the rebuttal. I have read the rebuttal and my evaluation stays the same.


Review 4

Summary and Contributions: This paper considers finding the parents of the variable of interest (a response variable Y) through the invariant causal prediction (ICP) principle. Often the given data from multiple environments are not suitable to pinpoint what the parents are. Hence, to achieve the goal, it is desired to obtain data from different environments through interventions. The authors define intervention-stable sets and explore its properties to finally propose a few criteria to obtain new data for the active learning of the parents under ICP. Empirical results show the usefulness of such criteria compared to a baseline.

Strengths: The paper is simple(-looking) yet provides fundamental criteria of what can be done with ICP given the capability to obtain more data through active learning. The proposed criteria seem novel and results are sound. NeurIPS community recently embraces causality, and this paper, which is at the intersection of ML (prediction model) and causal inference, will be of interest to many members in the community.

Weaknesses: It may be inherent to ICP that there exists no unobserved confounders between two variables (e.g., the response variable and its parent). But it is hard to conceive the cases where one can strongly claim that there exist no unobserved confounders. L195 “For now, only single-variable interventions are considered.“ The whole point of active learning is getting a (near-)optimal number of interventions to achieve the goal (estimating the parents of the response variable). The sentence makes me think there will be a later section where intervening on multiple variables. It would be desirable to discuss at least what is challenging to work with multiple-variable interventions. How will it be different from limiting to single variable interventions? An atomic intervention is different from stochastic intervention since they can cut the incoming edges onto the intervened variables. For example, the example A.3 in the supplementary material will not include edges from X0 and X1 to X2 if the intervention is atomic. Then, {S2} and {S3} will be in the mathbbS_e and the intersection will not include {X0, X1}. Hence, one cannot just casually say that “do” or “different” types of interventions are feasible. (Further, “conditional” intervention is deterministic and the application of d-separation is more subtle.)

Correctness: I checked theorems and their proofs and found no specific problem except that intervening on a variable does not remove incoming edges onto it.

Clarity: The paper is written clearly in general. footnote 1 says that authors consider any type of intervention but it is not well specified. What is “a different type”? (also see the weaknesses regarding other types)

Relation to Prior Work: The paper summarized prior work well.

Reproducibility: Yes

Additional Feedback: I have read the authors' feedback. ================================= Thanks for a simple and elegant new work on active learning with ICP. Causal sufficiency can be clearly mentioned In L115–116, the authors mentioned that noise variables are jointly independent. If noise variables are unobserved variables as used in an SCM, jointly independence does not imply that Xs and Y are unconfounded. The causal sufficiency assumption seems more clear (to me). minor L319 One might ask whether L323 It would be interesting

[Author Response · NeurIPS 2020]

We would like to thank you for your time and valuable feedback. Thank you for helping us to improve our manuscript! We hope to have correctly understood your questions, and will try to exhaustively address all your comments.

Reviewer 4 questions the applicability of our theory for hard or atomic interventions. It is important to note that in our framework (Setting 1), interventions are represented as additional variables in the graph, not as edge deletions (also see section 3.2.2 of *Causality*[1]). This captures a wide range of interventions, including atomic interventions. The d-separation statement which defines intervention stable sets (Definition 2.1) is a statement made on this extended graph, instead of on the mutilated graph resulting from removing incoming edges to variables which receive atomic interventions. Loosely put, the idea is that stable sets block the flow of information between the response $Y$ and the exogenous variables whose distributions change with the environment, resulting in invariance (c.f. proof of Proposition 2). We agree to be more specific as to what we mean by "other types of interventions" in footnote 1, p. 3, and will change this to echo line 129. We thank reviewer 4 for the additional comments on the manuscript.

The proposed policies consider single-variable interventions (line 195): at each iteration, a sample is collected from an experiment where only one variable is intervened on. The theoretical results presented in section 2 also hold for multiple-variable interventions (line 129). Since the policies shrink the pool of possible targets and then pick at random, we could simply pick $k$ targets instead of one. We considered single-variable interventions for simplicity and to compare with ABCD, which also only considers single-variable interventions.

We agree that the base assumptions of ICP are strong; regarding their validity in practice, the method has shown competitive performance when applied to real gene expression data[2]. We agree with reviewer 4 that it is difficult to claim that there are no confounders between the target and its parents. Section 5 in ICP [27] outlines ideas for generalizations of the method in the presence of hidden confounders; these were materialized in the causal Dantzig[3] method. Combining this method with A-ICP is interesting future work. In any case, section 6.3 of the ICP paper considers the behaviour of the method under model violations (such as hidden confounders), with the result that these often do not have severe consequences for the coverage guarantees of ICP (albeit result in a loss of power).

Reviewer 2 asks if it is possible to directly intervene on the response variable to identify the parents. We are not entirely sure what the reviewer has in mind. It is true that if the Markov equivalence class is known, one could intervene on the response variable and perform conditional independence testing to orient all its surrounding edges. However, this would limit the interventions to be do (hard, atomic) interventions. Furthermore, estimating the Markov equivalence class is difficult with a finite sample (see related work), even more so when only a small observational sample is available, as we allow for our method. Another approach would be to intervene on the response and check which variables undergo a change in their marginal distributions, i.e. directly testing for a causal effect. However, this would only partition the predictors into variables which are downstream of the response and not. We hope this answers the reviewer's question.

Increasing the number of predictors (i.e. graph size) had the effect of lowering the relative performance of the random strategy, as its chances of picking a parent were reduced. Otherwise, results were very similar and we decided to not present them separately. An interesting question is how different signal-to-noise ratios affect the performance of the policies. To partially answer this question, Figure C.11 illustrates their performance for different intervention strengths.

We will provide a detailed analysis of the computational complexity in the final version. In any case, it is exponential in the number of predictors, which limits the applicability of our method to "large $p$" settings. However, we envision settings where the time needed to carry out an experiment far outweighs the computation time to select the next experiment, which is common in empirical sciences like biology. Thus, A-ICP could still be useful in settings with a moderate number of variables. We would like to note that in our experiments, the time required to run a single iteration of ABCD far exceeded that of A-ICP, due to the approximation of the posterior.

We believe that ABCD and A-ICP are complimentary in many ways, not only regarding their input requirements. We agree with reviewer 1 that, in terms of Jaccard similarity, ABCD has a better performance in the first few iterations, and would be the preferred method if this was the goal. On the other hand, if false positives incur a high cost, the more conservative A-ICP would be preferable, as it offers a strict control over the family-wise error rate.

Finally, we would like to stress that A-ICP is directly applicable to the nonlinear extension of ICP [14], as the theoretical results used to construct the policies make no assumption of linearity (lines 119, 322-325). We did not show results for nonlinear models for two reasons. First, the performance of nonlinear ICP is not great even for simple graphs; we feared this would confound the performance of the policies, whose evaluation was our main goal. Second, and along the same line, the Markov blanket estimation (for the policies that use it) is not as straightforward for nonlinear systems as it is in the linear case.

## Footnotes

[1] Pearl, J., 2009. Causality. Cambridge university press. [2] Meinshausen, N., Hauser, A., Mooij, J.M., Peters, J., Versteeg, P. and Bühlmann, P., 2016. Methods for causal inference from gene perturbation experiments and validation. PNAS, 113(27), pp.7361-7368.

[3] Rothenhäusler, D., Bühlmann, P. and Meinshausen, N., 2019. Causal Dantzig: Fast inference in linear structural equation models with hidden variables under additive interventions. The Annals of Statistics, 47(3), pp.1688-1722.


[Meta-Review · NeurIPS 2020]

The authors propose an active causal learning framework that extends invariant causal prediction (ICP) to the active online setting. ICP assumes that the conditional distribution of the response, given its direct causes, remains invariant when intervening on arbitrary variables in the system. Several active learning policies for performing interventions and recovering the parent set of the target variable are also proposed. Overall the reviewers liked the paper but they have some reservations about the computational complexity of the proposed approach.